# Modular access to alkylfluorides via radical decarboxylative-desulfonylative *gem*-difunctionalization

Xianjin Wang[1,3], Haotian Li[2,3], Yasu Chen[1], Ziqiang Wang[1,2], Xinxin Wu[2] & Chen Zhu [1,2] ✉

Fluorine-containing compounds hold pivotal importance in life sciences. Recent decades have witnessed significant research efforts toward developing practical fluorination methods. Radical-mediated decarboxylative fluorination has proven to be a robust approach for incorporating diverse monofluoroalkyl groups. Here we show a radical-mediated modular synthesis of alkyl fluorides through a decarboxylative-desulfonylative *gem*-difunctionalization under mild photochemical conditions. The multi-component reaction proceeds in a controlled sequence of radical decarboxylation and heteroaryl migration, governed by radical polarity and kinetic effects, resulting in a wide range of valuable alkyl fluorides. Two C-C bonds and one C-F bond are concurrently formed throughout the process. Both styrenes and aliphatic alkenes serve as suitable substrates for this transformation. Furthermore, this method can be applied to the incorporation of a monofluoroalkyl moiety into complex alkene molecules at a late stage.

The integration of a fluorine atom or fluorinated groups into the molecular frameworks of pharmaceuticals, agrochemicals, and organic materials frequently elicits substantial enhancements in their physical, chemical, and biological functionalities, including improvements in lipophilicity, permeability, polarity, and metabolic resilience[1–4]. Owing to such profound impacts, fluorine-containing compounds are pivotal in fields related to life sciences, with estimates suggesting that over 20% of pharmaceuticals on the market today are fluoro-pharmaceuticals[5–8]. Consequently, the past decades have witnessed an intense focus on the development of versatile and efficient synthetic methods for fluorination[3,9–15]. In this vein, radical-mediated decarboxylative fluorination emerges as a distinguished approach, offering a gateway to a variety of monofluoroalkyl moieties prevalent in top-tier medications and an array of preclinical entities (Fig. 1a). Despite the remarkable evolution in reaction modalities that encompass traditional transition-metal catalysis inspired by Hunsdiecker-type reactions to modern photo-/electro-catalytic processes[16–25], the

shortcomings of these protocols (e.g., the limited reaction mode that constricts the diversity of products, and the poor availability of precursors required for complex product constructions) remain inadequately addressed (Fig.1b). In pursuit of expanding the chemical landscape, there is a high demand for preparing intricate alkyl fluoride compounds via multi-component decarboxylative fluorination, employing readily available alkenes as substrates.

We envision a modular pathway to synthesize alkyl fluorides via a multi-component reaction that involves $\alpha$-sulfonyl carboxylic acid **1**, alkene, and Selectfluor (Fig. 1c). This fluorination process unfolds through an orchestrated radical decarboxylation-desulfonylation cascade[26]. Presumably, the decarboxylation of **1** through single-electron transfer (SET) yields the intermediate **a**[27], which subsequently adds across the alkene to generate intermediate **b**. The following process involves the R[3] group migrating from the sulfone to the carbon-centered radical with a concurrent elimination of $SO_2$, leading to intermediate **c**[28–32]. The reaction with Selectfluor converts

[1]Frontiers Science Center for Transformative Molecules, School of Chemistry and Chemical Engineering, State Key Laboratory of Synergistic Chem-Bio Synthesis, and Shanghai Key Laboratory for Molecular Engineering of Chiral Drugs, Shanghai Jiao Tong University, Shanghai, China. [2]Key Laboratory of Organic Synthesis of Jiangsu Province, College of Chemistry, Chemical Engineering and Materials Science, Soochow University, Suzhou, Jiangsu, China. [3]These authors contributed equally: Xianjin Wang, Haotian Li. ✉e-mail: chzhu@sjtu.edu.cn

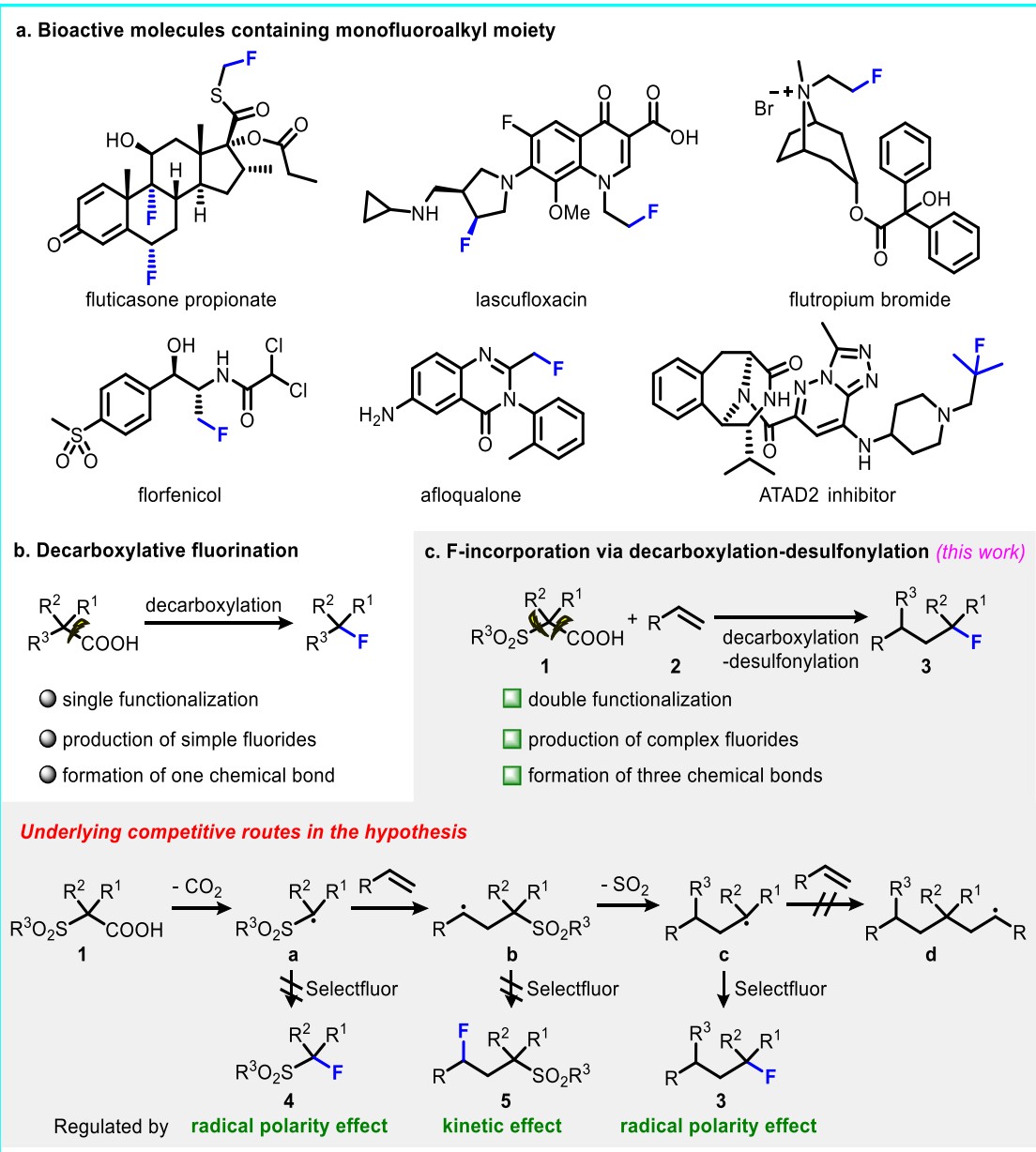

**Fig. 1 | The importance of monofluoroalkyl moiety and its incorporation by radical approaches. a** The importance of monofluoroalkyl moiety in bioactive molecules. **b** Classic decarboxylative fluorination. **c** Fluorination via decarboxylation-desulfonylation.

intermediate **c** into the desired product **3**. Theoretically, both radical intermediates **a** and **b** could potentially precede radical **c** in reacting with the fluorinating agent, thereby disrupting the formation of product **3**. Several considerations, however, offer insight into achieving the desired chemoselectivity: 1) The sulfone-associated alkyl radical **a** is electrophilic, making it less inclined to abstract a fluorine atom from Selectfluor due to mismatched polarity; 2) Alkyl radical **b** favors intramolecular migration of the $R^3$ group through a kinetically preferred five-membered cyclic transition state over intermolecular fluorine abstraction; and 3) The nucleophilic alkyl radical **c** is likely to engage in fluorine abstraction rather than addition to the electron-rich alkene, forestalling the formation of intermediate **d**. These rationales contribute to an anticipation of selectivity governed by radical polarities and kinetic influences. Nevertheless, the presence of competitive pathways still renders the overall transformation somewhat unpredictable.

Herein we report preliminary evidence in support of the hypothesis. The desired multi-component reaction proceeds under photochemical conditions, exhibiting remarkable chemoselectivity. Utilization of α-sulfonyl carboxylic acids provides double reacting sites compared to conventional fatty acids, facilitating the synthesis of an extensive spectrum of intricate alkyl fluorides. This process forges two C-C bonds and a C-F bond within the radical cascade of decarboxylation and desulfonylation.

## Results

### Reaction parameters survey

Optimization of the reaction parameters was pursued using styrene **1a** and α-sulfonyl carboxylic acid **2a** under photochemical conditions (Table 1). A thorough investigation revealed that employing Selectfluor as the fluorine donor, $KHCO_3$ as the base, and [Ir(dF(CF$_3$) ppy)$_2$(dtbbpy)]PF$_6$ as the photoredox catalyst in dichloromethane

## Table 1 | Reaction Optimization[a]

| Entry | Variation from the standard conditions | Yield (%)[b] |
|---|---|---|
| 1 | None | 63 |
| 2 | Fukuzumi dye instead of [Ir] | 27 |
| 3 | 4CzIPN instead of [Ir] | 36 |
| 4[c] | AgNO₃ instead of [Ir] | 20 |
| 5 | Cs₂CO₃ instead of KHCO₃ | 22 |
| 6 | K₂CO₃ instead of KHCO₃ | 28 |
| 7 | MeCN instead of DCM | 22 |
| 8 | DMF instead of DCM | 18 |
| 9 | THF instead of DCM | 25 |
| 10 | Dioxane instead of DCM | 15 |
| 11 | in dark | Trace |
| 12 | without [Ir] | Trace |
| 13 | without KHCO₃ | Trace |
| 14 | under air | Trace |

[a]Standard reaction conditions: **1a** (0.1 mmol), **2a** (0.2 mmol), [Ir(dF(CF₃)ppy)₂(dtbbpy)]PF₆ (2 mol %), Selectfluor (0.3 mmol), and KHCO₃ (0.3 mmol) in DCM (1 mL), irradiated with 30 W blue LEDs at 20 °C under N₂ for 48 h.
[b]Yield of isolated products.
[c]AgNO₃ (30 mol %) was used without light irradiation.

(DCM) yielded the targeted alkyl fluoride **3a** in a synthetically useful yield (Table 1, entry 1). Substitution of [Ir(dF(CF₃)ppy)₂(dtbbpy)]PF₆ with other frequently used photoredox catalysts, including Fukuzumi's salt (Mes·Acr⁺ClO₄⁻) and 4CzIPN, resulted in diminished yields of **3a** (Table 1, entries 2 and 3). Although possible, achieving **3a** via silver-catalyzed decarboxylative fluorination[19,33–35] proved less efficacious (Table 1, entry 4). Evaluation of various bases and solvents indicated that the replacement of KHCO₃ and DCM negatively affected the overall effectiveness of the reaction (Table 1, entries 5-10). Control experiments underscored the indispensability of light, catalyst, and base for this transformation, with exposure to air entirely inhibiting the conversion (Table 1, entries 11-14).

### Substrate scope

With the optimized reaction conditions in hand, we started to evaluate the generality of the protocol (Fig. 2). Various styrenes were initially tested, showing that the corresponding products were consistently produced (**3a**-**3n**), independent of the electronic characteristics and steric hindrance of the aryl substituents. Naphthyl and hetaryl such as thienyl ethylene are also suitable substrates (**3o** and **3p**). Both enamide and enol ether served as effective substrates, yielding the anticipated adducts (**3q** and **3r**). Notably, the protocol demonstrated robustness through its application to a range of unactivated aliphatic alkenes (**3s**-**3ag**). 1,1-Disubstituted alkenes, including low-boiling-point isobutene, efficiently formed all-carbon quaternary centers in the products (**3w** and **3x**). Many sensitive functional groups remained intact during the reaction, including alkyl bromide (**3aa**), epoxide (**3ab**), unprotected alcohol (**3ac**), and nitro group (**3ae**). A set of α-sulfonyl carboxylic acids was conveniently synthesized and then subjected to standard conditions. The sulfonyl part could be substituted with various heteroaryl groups. Beyond the parent benzothiazolyl group, functionalized benzothiazolyl, thiazolyl, quinolyl, pyridyl, and thienyl groups were readily integrated into the products (**3ah**-**3am**). Moreover, the aliphatic backbone of the carboxylic acids was diversified,

accommodating acyclic, cyclic, and even heterocyclic alkyl groups for decarboxylative coupling to alkenes (**3an**-**3ar**). However, secondary and primary carboxylic acids are so far unsuitable for the transformation, probably attributed to the low efficiency to generate relatively high-energy primary and secondary alkyl radicals via radical decarboxylation, as well as the lower stability of these radicals. Remarkably, this method was also applicable to the monofluoroalkylation of complex alkenes derived from natural products and drug molecules (**3as**-**3aw**).

### Synthetic applications

The products were proven to be versatile precursors to several valuable compounds, further underscoring the practicality of the synthetic approach (Fig. 3). The benzothiazolyl moiety in compound **3a** was efficiently transformed into a formyl group, yielding the fluorinated aldehyde **4**. Upon the treatment with diethylaminosulfur trifluoride (DAST), compound **4** was converted into the 1,1,4-trifluoroalkane **5**. A different sequence involving initial reduction of **4** with NaBH₄ and subsequent exposure to DAST generated the 1,4-difluoroalkane **6**. The oxidation of **4** via the Bayer-Villiger reaction with *meta*-chloroperoxybenzoic acid (*m*-CPBA) led to the formation of the formate **7**. Additionally, the reaction of **4** with a carbene generated from ethyl diazoacetate produced the fluorinated β-keto ester **8**.

To gain further insight into the reaction pathways, a set of mechanistic studies was carried out. There was no change in the yield under dark conditions and no change in the yield after the reaction time was extended to 48 hours, confirming the light-dependent nature of the reaction (Fig. 4a). Quantum yield measurements ($\Phi = 0.55$) further support the involvement of a photocatalytic pathway in the reaction (refer to 'Quantum Yield Measurements' in the Supplementary Information for detailed information). However, radical chain pathway could not be entirely ruled out from the transformation. Cyclic voltammetry revealed that the oxidation potential of the corresponding potassium salt of **2a** ($E_{p/2} = +1.49$ V vs

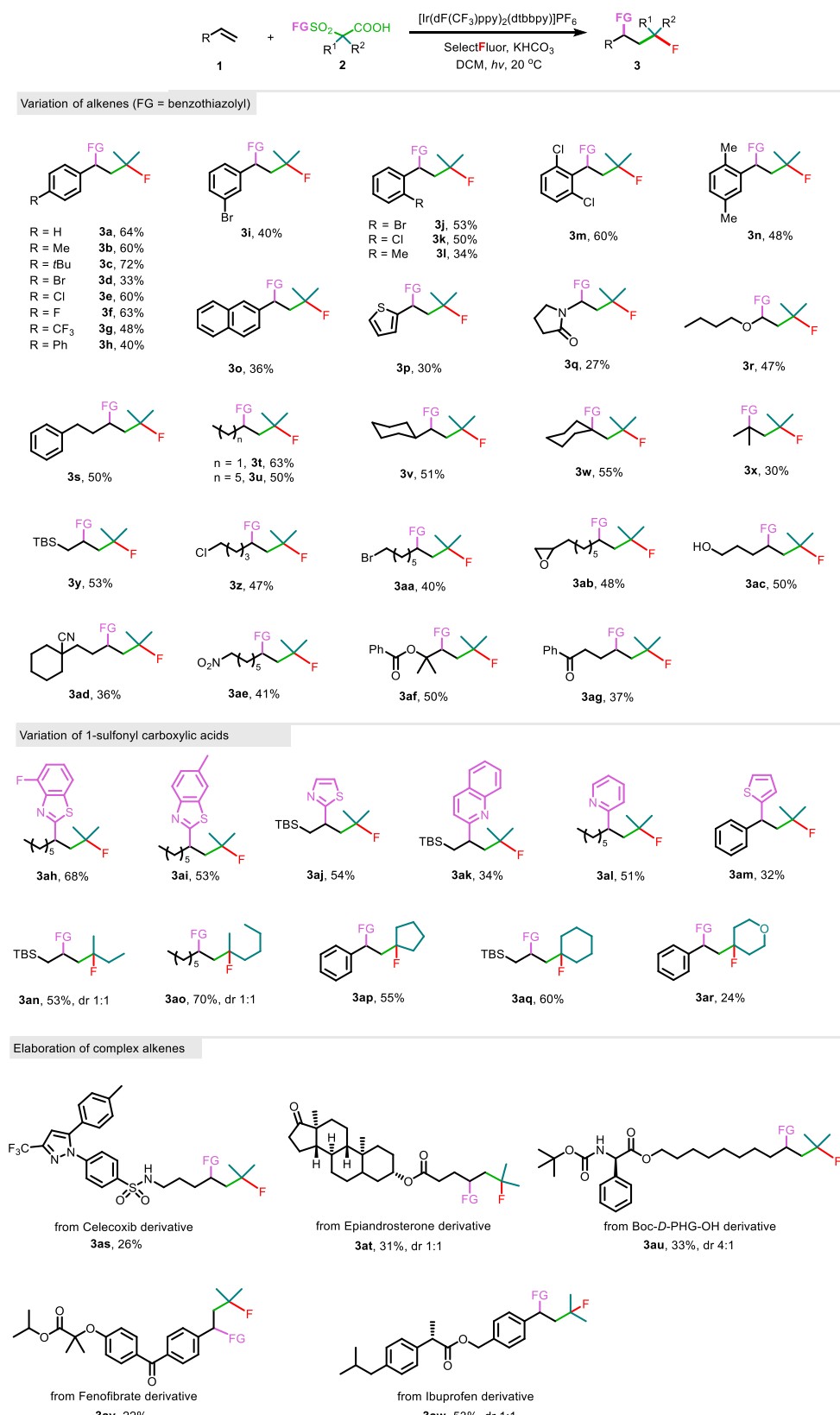

**Fig. 2 | Generality assessment of the reaction.** Reaction conditions: **1** (0.2 mmol), **2** (0.4 mmol), [Ir(dF(CF$_3$)ppy)$_2$(dtbbpy)]PF$_6$ (2 mol %), Selectfluor (0.6 mmol), and KHCO$_3$ (0.6 mmol) in DCM (2 mL), irradiated with 30 W blue LEDs at 20 °C under N$_2$ for 48 h.

SCE in MeCN) is substantially higher than that of Ir$^{IV}$ ($E_{1/2}^{III*/II}$ = +1.21 V vs SCE in MeCN)[36,37], ruling out the possibility of single-electron oxidative decarboxylation initiated by the excited Ir$^{III}$ species (Fig. 4b). This conclusion is bolstered by Stern-Volmer studies, where

the [Ir(dF(CF$_3$)ppy)$_2$(dtbbpy)]PF$_6$ was not reductively quenched by **2a**, but was quenched by Selectflour ($E_{1/2}^{red}$ = +0.33 V vs SCE in MeCN; $E_{1/2}^{III*/IV}$ = −0.89 V vs SCE in MeCN) (Fig. 4c)[38–40]. The notable decrease in the Stern-Volmer plot slope for **2a** was ascribed to the overlapping

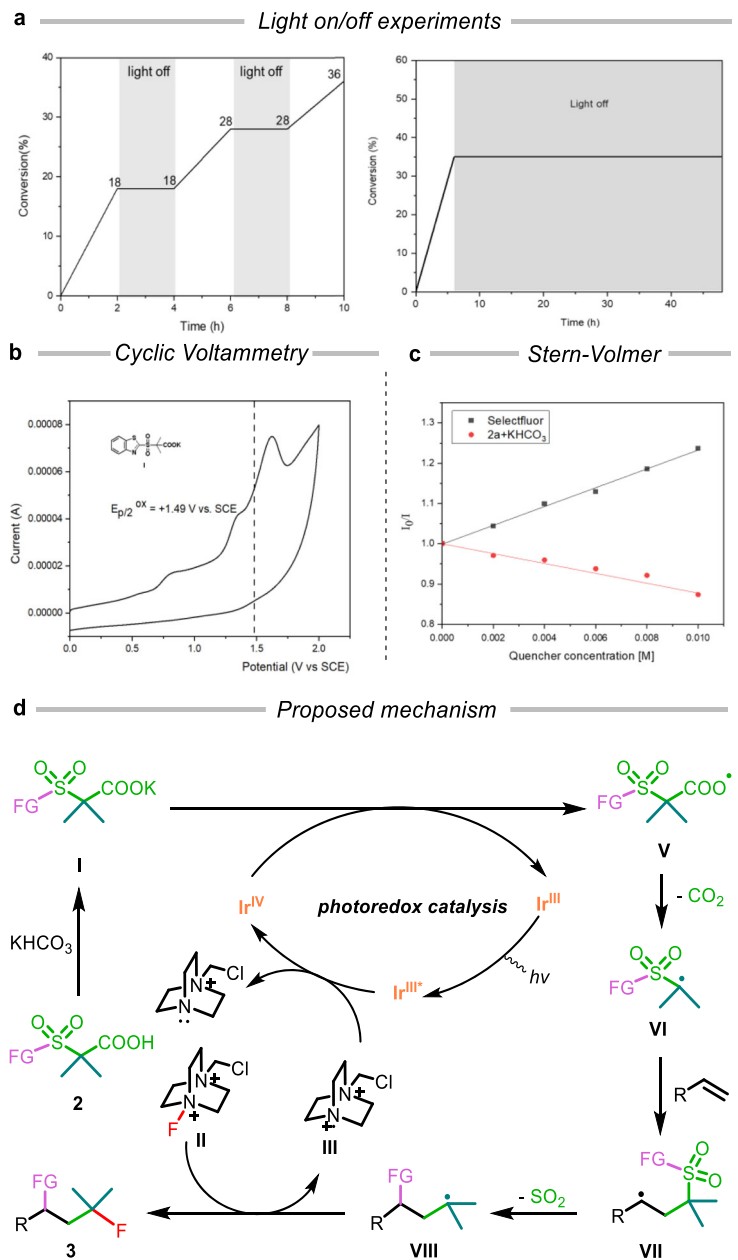

**Fig. 3 | Product transformations. a** The synthesis of **4**. Reaction conditions: (1) **3a** (0.1 mmol), 4 Å molecular sieves powders (150 mg), Me₃OBF₄ (0.5 mmol), CH₂Cl₂, 10 min; (2) MeOH (1 mL), 0 °C, NaBH₄ (0.25 mmol); (3) AgNO₃ (0.3 mmol), CH₃CN (1.5 mL)/H₂O(0.18 mL). Further transformations of **4**. Reaction conditions: **4** (0.1 mmol). **b** DAST (1.2 equiv), rt, 4 h; **c** Na₂HPO₄ (1.2 equiv), m-CPBA (3 equiv), 0 °C, 12 h; **d** (1) NaBH₄ (1.2 equiv), MeOH, rt, 4 h; (2) DAST (1.2 equiv), CH₂Cl₂, rt, 8 h; **e** MoO₂Cl₂ (5 mol %), N₂CHCOOEt (1.2 equiv), CH₂Cl₂, 30 °C. Isolated yields are shown.

**Fig. 4 | Mechanistic studies and proposed mechanism. a** Light on/off experiment. **b** Cyclic voltammogram of the potassium salt of **2a** in MeCN. **c** Stern-Volmer Studies. **d** Proposed reaction mechanism.

light absorption by **2a**, which caused an increase in fluorescence intensity.

## Proposed mechanism

The proposed mechanism is illustrated in Fig. 4d. Initially, exposing [Ir(dF(CF$_3$)ppy)$_2$(dtbbpy)]PF$_6$ to visible light excites it to the Ir$^{III*}$ state, which then engages in the oxidative quenching with *N*-radical cation (**III**) to generate Ir$^{IV}$ species. A single-electron transfer (SET) from Ir$^{IV}$ to the carboxylate anion (**I**)-formed by deprotonating carboxylic acid (**2**)-yields the carboxy radical (**V**). This radical rapidly undergoes decarboxylation, producing the alkyl radical (**VI**). The alkyl radical (**VI**) then adds to the alkene, initiating an intramolecular heteroaryl migration followed by SO$_2$ expulsion. The resultant alkyl radical (**VIII**) abstracts a fluorine atom from Selectfluor (**II**) to create the targeted product (**3**), simultaneously regenerating the radical cation (**III**).

## Discussion

We present a modular synthesis of alkylfluorides through a sequence of radical decarboxylation-desulfonylation reactions. Departing from conventional radical decarboxylative fluorination approaches, this protocol constructs multiple chemical bonds simultaneously and introduces a versatile means to assemble alkylfluorides with remarkable structural intricacy and extensive diversity. The multi-component reaction proceeds orderly under mild photochemical conditions, steered by the interplay of radical polarity and kinetic influences. Both styrene derivatives and the more challenging non-activated alkenes, particularly aliphatic alkenes, are suitable substrates for this transformation. Furthermore, this method allows for the incorporation of a monofluoroalkyl moiety into complex alkene frameworks.

## Methods

### General procedure for the synthesis of alkyl fluorides

**1** (0.2 mmol), **2** (0.4 mmol), [Ir(dF(CF$_3$)ppy)$_2$(dtbbpy)]PF$_6$ (2 mol %), Selectfluor (0.6 mmol), and KHCO$_3$ (0.6 mmol) in DCM (2 mL) was loaded in a 4 mL reaction vial. The mixture was stirred with 30 W blue LEDs at 20 °C under N$_2$ for 48 h. After evaporation of the solvent under reduced pressure, the residue was purified by flash column chromatography on silica gel to give the desired product.

## Data availability

The authors declare that all other data supporting the findings of this study are available within the article and Supplementary Information files, and also are available from the corresponding author on request.

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

## Acknowledgements
The authors are grateful for the financial support from the National Natural Science Foundation of China (grant numbers 22171201 and 22371185, C.Z.), the Fundamental Research Funds for the Central Universities (23×010301599 and 24×010301678, C.Z.), and the Program of Shanghai Academic/ Technology Research Leader (23XD1421900, C.Z.).

## Author contributions
C.Z. conceived the concept and directed the project. X.-J.W., H.L., Y.C., and Z.W. performed experiments. X.-J.W., H.L., and X.-X.W. prepared the Supplementary Information. C.Z. and X.-X.W. wrote the paper. All authors discussed the results and commented on the manuscript.

## Competing interests
The authors declare no competing interests.
