## [Peer Review file · Nature Communications]

Modular Access to Alkylfluorides via Radical Decarboxylative-Desulfonylative gem-Difunctionalization

Corresponding Author: Professor Chen Zhu

Version 0:

Reviewer comments:

Reviewer #1

(Remarks to the Author)

This manuscript presents a novel approach for the synthesis of alkyl fluorides through a radical decarboxylative-desulfonylative gem-difunctionalization under mild photochemical conditions. The method involves a multi-component reaction utilizing α -sulfonyl carboxylic acid, an alkene, and Selectfluor as reagents. The authors detail their optimization efforts, substrate scope, and mechanistic insights into this transformation. The protocol demonstrates broad applicability across various substrates including styrenes, aliphatic alkenes, and complex natural products/drug molecules, highlighting its versatility and potential utility in medicinal chemistry.

Overall, the manuscript represents a significant contribution to the field of fluorine chemistry and organic synthesis, offering a new strategy for the preparation of alkyl fluorides. With minor revisions addressing the points mentioned below, I believe this work is suitable for publication.

1. The carboxylic acid substrates in this study appear to be limited to tertiary carboxylic acids. Have other types of carboxylic acids (e.g., primary or secondary) been attempted? If so, please provide the results and discussion; if not, explain why this method is particularly suited for tertiary carboxylic acids and discuss any potential limitations. This would help readers better understand the scope and possible restrictions of this methodology.
2. When discussing decarboxylative fluorination and Smiles rearrangement during decarboxylation, it is necessary to cite recent relevant literature to showcase the latest advancements in the field. Specifically, the following references should be included: Chem Catalysis 4, 101162; Org. Lett. 2024, 26, 50, 10940–10945.
3. The manuscript mentions that multiple byproducts are avoided through polarity matching and kinetic advantages, which is a cleverly designed strategy. However, during the extensive substrate scope investigation, were any byproducts observed? Additionally, the paper notes that carbon dioxide (CO₂) and sulfur dioxide (SO₂) are generated during the reaction. Can these gases be detected experimentally? Techniques such as gas chromatography (GC) or infrared spectroscopy (IR) could be used to verify their formation. Regarding the free radical intermediates, is it possible to observe or capture these intermediates through experiments like trapping studies or other methods?

Reviewer #2

(Remarks to the Author)
see the attached files.

Version 1:

Reviewer comments:

Reviewer #1

(Remarks to the Author)

The author has effectively addressed the mentioned questions, particularly with the new experiments that strongly support the mechanism and the author's viewpoint. The revised manuscript is suitable for publication.

Reviewer #2

(Remarks to the Author)

The authors have responded to all of my comments, presenting new data and well - reasoned explanations. As a result, the revised manuscript has shown significant improvement and is now of much higher quality. Therefore, I wholeheartedly recommend the acceptance of this revised manuscript.

Dear Editor,

Thanks for your kind decision. The manuscript has been revised in accordance with the comments from both referees. We provide a detailed point-by-point response below. We hope you will be satisfied with this revised manuscript.

Response to Reviewer # 1

1. The carboxylic acid substrates in this study appear to be limited to tertiary carboxylic acids. Have other types of carboxylic acids (e.g., primary or secondary) been attempted? If so, please provide the results and discussion; if not, explain why this method is particularly suited for tertiary carboxylic acids and discuss any potential limitations. This would help readers better understand the scope and possible restrictions of this methodology.

Response: Thanks for the suggestion. We have synthesized primary and secondary carboxylic acids and attempted the reaction under the standard conditions. Unfortunately, however, no target products were obtained. It could be attributed to the low efficiency to generate relatively high-energy primary and secondary alkyl radicals via radical decarboxylation, as well as the lower stability of these radicals. The discussion has been added in Page 6.

2. When discussing decarboxylative fluorination and Smiles rearrangement during decarboxylation, it is necessary to cite recent relevant literature to showcase the latest advancements in the field. Specifically, the following references should be included: *Chem Catalysis* 4, 101162; *Org. Lett.* 2024, 26, 50, 10940-10945.

Response: Thank you for the suggestions. The suggested two papers have been added as ref. 16 and 28.

3. The manuscript mentions that multiple byproducts are avoided through polarity matching and kinetic advantages, which is a cleverly designed strategy. However, during the extensive substrate scope investigation, were any byproducts observed? Additionally, the paper notes that carbon dioxide (CO₂) and sulfur dioxide (SO₂) are generated during the reaction. Can these gases be detected experimentally? Techniques such as gas chromatography (GC) or infrared spectroscopy (IR) could be used to verify their formation. Regarding the free radical intermediates, is it possible to observe or capture these intermediates through experiments like trapping studies or other methods?

Response: We sincerely thank the reviewer for the comment. The existence of byproduct was observed and identified by ¹H-NMR (see below). We employed TEMPO to capture radical intermediates during the reaction and detected the formation of TEMPO-benzothiazolyl adduct.

¹H-NMR spectra (400 MHz, CDCl₃, 25 °C) of byproduct

HRMS result

Additionally, the generation of CO₂ could be detected by GC-MS.

CO₂ detected by GC-MS

Upon analyzing the reaction, we concluded that the DABCO-byproduct generated from Selectfluor might capture SO₂, leading to the formation of a zwitterion complex (see below). Indeed, such complex could be identified by HRMS, supporting the formation of SO₂ in the reaction.

HRMS result

Response to Reviewer #2

1. Formatting Inconsistencies: Ensure that the R³ in Figure 1c is presented in the same color.

Response: Done, we have modified Figure 1c.

2. Migration of Phenyl Ring: In this reaction, all the migrated units are heterocycles. It would be interesting to investigate whether a phenyl ring could migrate in this cascade transformation.

Response: Thanks for the suggestion. In fact, we have tried phenyl migration, but did not get any products in the reaction. It is reasonable as, to our experience, the phenyl migration is much more difficult than heteroaryls.

3. Formation of Cyano-Migrated Product: Regarding the formation of compound **3ad**, did the authors observe the formation of the cyano-migrated product? Since this process also appears to be kinetically favorable, it is worth exploring.

Response: Thanks for the suggestion. We carefully analyzed the reaction of **3ad** by subjecting the reaction mixture to GC-MS analysis and NMR analysis. In this reaction, the cyano-migrated byproduct was not detected, probably due to the Smiles rearrangement by cleaving the C-SO₂ bond is more favorable than the C-C bond for cyano migration.

4. Reasons for Low Yields: For the relatively low yields of the majority of products, is it due to low conversion or the formation of other side products? The authors are expected to clarify this in the manuscript.

Response: Thanks for the suggestion. We observed that the light transmittance is

gradually decreased during the reaction, due to the presence of insoluble KHCO_3 , which would impede the conversion. Additionally, the formation of byproducts was detected (see below). We hypothesize that the diminished light transmittance, coupled with the generation of byproducts, resulted in the decreased yields.

$^1\text{H-NMR}$ spectra (400 MHz, CDCl_3 , 25 °C) of byproduct

5. Interpretation of Cyclic Voltammetry Peaks: In the interpretation of cyclic voltammetry, what do the other two peaks around 0.8 V and 1.25 V represent?

Response: Thank you for the suggestions. In the cyclic voltammetry analysis, the other two peaks observed at around 0.8 V and 1.25 V were caused by the electrodes or components within the instrument during testing. We retested of the cyclic voltammogram in MeCN. It was observed that in the blank control (without adding **2a**), the two peaks at around 0.8 V and 1.25 V were also appeared. Despite repeated efforts to clean the electrode thoroughly, the peaks persisted at these positions.

Blank control

Test result

6. Incorrect ^{13}C -NMR Data Interpretation: The ^{13}C -NMR data for compound **3g** was misinterpreted. The C-F coupling of the CF_3 group in the phenyl ring should be around 275 Hz instead of the reported 240.9 Hz. The authors are required to double-check and revise all the data for the fluorinated compounds.

Response: We have revised the SI, and rechecked all of the data thoroughly.

Comment to the authors NCOMMS-25-03939

Zhu and co-workers reported a radical-mediated modular synthesis of alkyl fluorides via a cascade process involving radical addition, functional group migration, and electrophilic fluorination under mild photocatalysis conditions. In this process, three new bonds were simultaneously formed, and a wide variety of alkenes and styrenes were compatible. This method represents a further advancement in their consistent research interest in the radical docking-migration strategy for the modular derivatization of feedstock chemicals. The manuscript is well-organized, and the methods introduced are likely to be of interest to the scientific community. The topic is highly relevant and timely, mirroring the current trend towards sustainable and photochemical synthetic methodologies. Once the following issues are satisfactorily addressed, I strongly recommend this manuscript for publication in Nature Communications.

- 1. Formatting Inconsistencies:** Ensure that the R^3 in Figure 1c is presented in the same color.
- 2. Migration of Phenyl Ring:** In this reaction, all the migrated units are heterocycles. It would be interesting to investigate whether a phenyl ring could migrate in this cascade transformation.
- 3. Formation of Cyano - Migrated Product:** Regarding the formation of compound **3ad**, did the authors observe the formation of the cyano-migrated product? Since this process also appears to be kinetically favorable, it is worth exploring.
- 4. Reasons for Low Yields:** For the relatively low yields of the majority of products, is it due to low conversion or the formation of other side products? The authors are expected to clarify this in the manuscript.
- 5. Interpretation of Cyclic Voltammetry Peaks:** In the interpretation of cyclic voltammetry, what do the other two peaks around 0.8 V and 1.25 V represent?

6. Incorrect ^{13}C - NMR Data Interpretation: The ^{13}C - NMR data for compound **3g** was misinterpreted. The C - F coupling of the CF_3 group in the phenyl ring should be around 275 Hz instead of the reported 240.9 Hz. The authors are required to double - check and revise all the data for the fluorinated compounds.